# Transforming Recurrent Neural Networks with Attention and Fixed-point Equations

## Abstract

Transformer has achieved state of the art performance in multiple Natural Language Processing tasks recently. Yet the Feed Forward Network(FFN) in a Transformer block is computationally expensive. In this paper, we present a framework to transform Recurrent Neural Networks(RNNs) and their variants into self-attention-style models, with an approximation of Banach Fixed-point Theorem. Within this framework, we propose a new model, StarSaber, by solving a set of equations obtained from RNN with Fixed-point Theorem and further approximate it with a Multi-layer Perceptron. It provides a view of stacking layers. StarSaber achieves better performance than both the vanilla Transformer and an improved version called ReZero on three datasets and is more computationally efficient, due to the reduction of Transformer's FFN layer. It has two major parts. One is a way to encode position information with two different matrices. For every position in a sequence, we have a matrix operating on positions before it and another matrix operating on positions after it. The other is the introduction of direct paths from the input layer to the rest of layers. Ablation studies show the effectiveness of these two parts. We additionally show that other RNN variants such as RNNs with gates can also be transformed in the same way, outperforming the two kinds of Transformers as well.

## 1 Introduction

Recurrent Neural Network, known as RNN, has been widely applied to various tasks in the last decade, such as Neural Machine Translation (Kalchbrenner & Blunsom, 2013; Sutskever et al., 2014), Text Classification (Zhou et al., 2016), Name Entity Recognition (Zhang & Yang, 2018; Chiu & Nichols, 2016), Machine Reading Comprehension (Hermann et al., 2015; Kadlec et al., 2016) and Natural Language Inference (Chen et al., 2017; Wang et al., 2017). Models applied to these tasks are not the vanilla RNNs but two of their famous variants, Gated Recurrent Unit (Cho et al., 2014), known as GRU, and Long Short Term Memory (Hochreiter & Schmidhuber, 1997), known as LSTM, in which gates play an important role. RNNs are hard to be computed parallelly. They are not bidirectional either, meaning that a word cannot utilize the information of words coming after it. A general way to alleviate this problem is to reverse the input sequence and combine results given by two different RNN encoders with operations like concatenation and addition.

However, Transformer (Vaswani et al., 2017) has provided a better solution. It is based on purely attention mechanism, which has been widely used in Neural Machine Translation since Bahdanau et al. (2014). Models based on self-attention mechanism are mostly Transformer and its variants, such as Transformer-XL (Dai et al., 2019), Universal Transformer (Dehghani et al., 2019) and Star-Transformer (Guo et al., 2019). Compared with recurrent units such as GRU and LSTM, self-attention-style models can be computed parallelly, which means they suit better large-scale training. But each of these Transformers has an FFN layer with a very high vector dimension, which still is the bottleneck to improve the computation efficency.

In this paper, we present a new framework based on Banach Fixed-point Theorem to transform the vanilla RNN and its variants with self-attention mechanism. StarSaber, one of such transformed models, outperforms both the vanilla Transformer and ReZero (Bachlechner et al., 2020) in our experiments with less parameters and thus less computational power. To start with,

we need a different view of attention. Attention is a way to build a relation graph between words, and the vanilla RNN is nothing but a model with a relation graph as a chain. This graph is in fact represented with an adjacent matrix, which is computed by mapping each pair of positions to a positive real number and normalizing the numbers related to each position, which are just those in the same row of the adjacent matrix, so that they sum up to one.

The vanilla RNN updates hidden states through a chain, that is, the hidden state for each position only depends on that in the previous position. However, if we have this relation graph, the hidden state for each position depends on hidden states for all other positions in a sequence. This is where we obtain equations. In our opinion, a bidirectional RNN is defined by some equations and Banach Fixed-point Theorem inspires us to iterate according to them. When we fix the number of iterations and specify distinct weights for each of them, a self-attention-style model is then constructed.

In Transformer, Position Embedding(PE) as a way to capture word order information in language by adding a matrix to the input, is indispensable. But in StarSaber, position encoding is done in the aggregation step after the construction of a relation graph. For each position, we sum up linear transformations of hidden states in all positions with the corresponding weights in the relation matrix in order to get an attention vector. In the calculation of such a vector, we specify different linear transformation weights for the "future" and the "past". Then the hidden vector for a position is computed with the corresponding attention vector and an input vector, which turns into a direct path from the input layer to each hidden layer. And we directly drop the FFN layer in Transformer achieving still competive and even better results with much less parameters on three datasets provided by CLUE (Xu et al., 2020): the AFQMC dataset of Sentence Similarity, the TNEWS dataset of Text Classification and the CMNLI dataset of Natural Language Inference. More importantly, our derivation of StarSaber shows a universal way to transform different RNNs, such as LSTM and GRU discussed in the following content, providing possibilities other than Transformers for self-attention models.

## 2 RELATED WORK

Gates were first introduced into recurrent networks in LSTM and were rediscovered and simplified in GRU. Gate mechanism is an operation that multiplies an output by a single sigmoid layer of the input and is often seen as an approach to address the gradient vanishing issue. But if only so, other approaches which addresses this problem should achieve similar results to LSTM and GRU. In this paper, we show by experiments that in StarSaber which doesn't have such a problem, gates can also help improve the performance.

Attention in sequence modeling is a weighted sum of the output in each position of a sequence, which simulates the way a man distributes his attention to all its parts. Weights in this sum are given by a certain function of some inputs. And self-attention is an approach computing both the weighted sum and weights on the same sequence without any other inputs. There are different types of attention like multi-head attention and scaled dot product attention in Transformer, attention based on addition in Bahdanau et al. (2014), and bilinear attention in Luong et al. (2015). Our model applies the bilinear attention in the construction of a word relation graph.

Residual Connection was proposed by He et al. (2015). It alleviates the problem of training deep neural networks. In Natural Language Processing, Residual Connection alleviates both the gradient vanishing problem and the degration problem of deep networks. Our model uses a weighted residual connection (Bachlechner et al., 2020) which further alleviates the degration problem. Another similar idea is the highway connection (Srivastava et al., 2015). In this paper, we inspect the gate mechanism in our self-attention-style model. Note that the highway connection can also fit into our framework, which is a fixed-point generalization of GRU.

Pretraining has proved to be extremely useful since Embeddings from Language Models(ELMO) (Peters et al., 2018). Many works that follow such as BERT (Devlin et al., 2018), ALBERT (Lan et al., 2020), XLNET (Yang et al., 2019) have outperformed humans. Pretraining is a training pattern which trains a language model, usually extremely large, on an enormous dataset with one

or more unsupervised tasks and fine-tunes it on other datasets and tasks. There are two types of language models in general, known as auto-regressive models(e.g., XLNET) and auto-encoder ones(e.g., BERT). However, pretraining on a large dataset requires resources. We show in this paper that only pretraining on a dadtaset formed by collecting the training, development and test inputs together can as well improve the performance, revealing the significance of pretraining tasks.

MLM is the unsupervised task utilized by BERT to pretrain. It randomly masks some pieces of a sentence and train the model to predict what has been masked. In this way, knowledge is gained and the model is initialized for downstream tasks. However, experiments show that even not pretrained on a large dataset, using MLM to pretrain on a dataset formed by collecting the training, development and test inputs can still improve the performance, inspiring us that a more flexible and task-related pretraining method is beneficial.

## 3 MODEL ARCHITECTURE

### 3.1 RECURRENT NEURAL NETWORKS AND BIDIRECTIONAL EQUATIONS WITH SELF-ATTENTION

This section follows the intuition we have discussed before. The vanilla RNN formulas are listed below:

$$h_n = \tanh(Uh_{n-1} + Wx_n) \tag{1}$$

In self-attention, we don't just utilize the hidden state from the previous position but hidden states from all positions, to compute a hidden vector for position n. To encode information of relative order, we specify distinct linear transformation weights. For simplicity, we ignore all bias terms in the following derivation. Following the bilinear self-attention mechanism, we have:

$$
\begin{aligned}
h_n &= \tanh(A_n + Vx_n) \\
A_n &= \sum_{i<n} G_{ni}U^{left}h_i + \sum_{i\geq n} G_{ni}U^{right}h_i{}^1 \\
G_{ni} &= \mathrm{softmax}(g, -1) = \frac{g_{ni}}{\sum_j g_{nj}} \\
g_{ni} &= \exp(\frac{h_n^T W h_i}{\sqrt{d}})^2
\end{aligned}
\tag{2}
$$

What we have done here is replacing $h_{n-1}$ with an attention vector $A_n$. Notice a fact that in the first equality, h appears on both the left-hand side and the right-hand side(we use h to compute $A_n$), turning it into an equation. This means a bidirectional model is defined by a set of equations, because the word relation graph constructed by attention is not free of loops. Moreover, introducing equations can be seen as a constraint to obtain stable representation of a sentence. Intuitively, if we view the non-linear function on the right hand side as an updating operation and the hidden vector we obtain in each position as a semantic representation, it simply means that when the model "reads" the whole sentence again based on the current understanding, it should produce the same representation, meaning that it has "fully understands" the whole sentence and makes no changes on the hidden vectors.

### 3.2 GENERALIZE EQUATIONS WITH FIXED-POINT

Now we have an equation to solve, which is extremely complex and difficult. But Banach Fixed-point theorem shows us a way.

**Theorem 3.1 (Banach Fixed-point Theorem)** *For any real-valued function f(x), if $|\frac{df}{dx}| < 1$, then iteration process $x_{n+1} = f(x_n)$ converges and $\lim_{n\to+\infty} x_n = x^*$, where $x^* = f(x^*)$.*

---

[1] If not stated, a sum without a limit is to sum over all possible values.

[2] All these U, V, Ws are matrices that satisfy rules of the matrix-vector product. The hyperparameter d here is the hidden size. The attention here is scaled for faster convergence.

The equation above is an equation of an iterative pattern, and this Theorem just tells us that as long as we keep iterating, we will obtain a root of the equation if its jacobian matrix satisfies some conditions. The iterative pattern is given as follows:

$$h_n^{l+1} = tanh(A_n^l + V x_n)$$
$$A_n^l = \sum_{i<n} G_{ni}^l U^{left} h_i^l + \sum_{i \geq n} G_{ni}^l U^{right} h_i^l$$
$$G_{ni}^l = softmax(g^l, -1) = \frac{g_{ni}^l}{\sum_j g_{nj}^l} \qquad (3)$$
$$g_{ni}^l = \exp(\frac{(h_n^l)^T W h_i^l}{\sqrt{d}})$$

We can then iterate till it converges. Similar ideas are in Bai et al. (2019), where the authors solve the fixed-point directly with very high computational cost. Sometimes it cannot even converge to a fixed-point, since the convergence condition is quite strict. A sufficient condition for convergence is that all parameter matrices are strictly orthogonal, making the optimization problem hard. Therefore, if we want to obtain a faster and more stable model, we can approximate it with a Multi-layer Perceptron(MLP) and relax the condition of convergence. In addition, we allow our model to assign different weights for different layers. The reason why we don't reuse parameters in each layer is that iterating with the same set of parameters without a constraint of orthogonality often diverges. Even if we fix the number of iterations, it is still hardly possible to converge to the correct fixed-point. In this case, specifying different weights for each layer allows our model to learn a better fit for the whole iteration process. Therefore, we have

$$h_n^{l+1} = tanh(A_n^l + V^l x_n)$$
$$A_n^l = \sum_{i<n} G_{ni}^l U^l h_i^l + \sum_{i \geq n} G_{ni}^l Q^l h_i^l$$
$$G_{ni}^l = softmax(g^l, -1) = \frac{g_{ni}^l}{\sum_j g_{nj}^l} \qquad (4)$$
$$g_{ni}^l = \exp(\frac{(h_n^l)^T W^l h_i^l}{\sqrt{d}})$$

Here we also need an initial point to start the iteration. In our model, we choose the input sequence itself to be the initial value, that is to set $h_i^0 = x_i$. In more general cases, the initial value may be a linear transformation of the input or just some fixed vector like a zero one.

### 3.3 RESIDUAL CONNECTIONS

Since we decide to approximate the iteration process with an MLP, Residual Connection is then indispensable in for it helps to alleviate the problem of degration. However, its magnitude, which is the fixed scaling number, needs to be tuned mannually. If we allow it to be automatically tuned by our model, the whole model can be written as follows:

$$h_n^{l+1} = h_n^l + \alpha^l tanh(A_n^l + V^l x_n) \qquad (5)$$

The rest of formulas are the same as above. The $\alpha^l$ here is a crucial weight initialized to be one or zero in every layer. In Bachlechner et al. (2020) it is initialized to be zero in order to stabilize Transformer. But in our experiments we don't train extremely deep networks with a thousand or more layers. Thus we initialize it to be one since we find that it speeds up convergence.

### 3.4 MODEL SUMMARY

The derivation above has demonstrated how to transform the vanilla RNN into a self-attention-style model. To summarize, the structure of StarSaber can be described by Figure 1. The Attention Graph here is the relation construction process returning a matrix G. It is exactly what happens in the last two formulas shown above. Masked Attention here is how we implement Position Encoding. And $\alpha$ is the weight for Residual Connection.

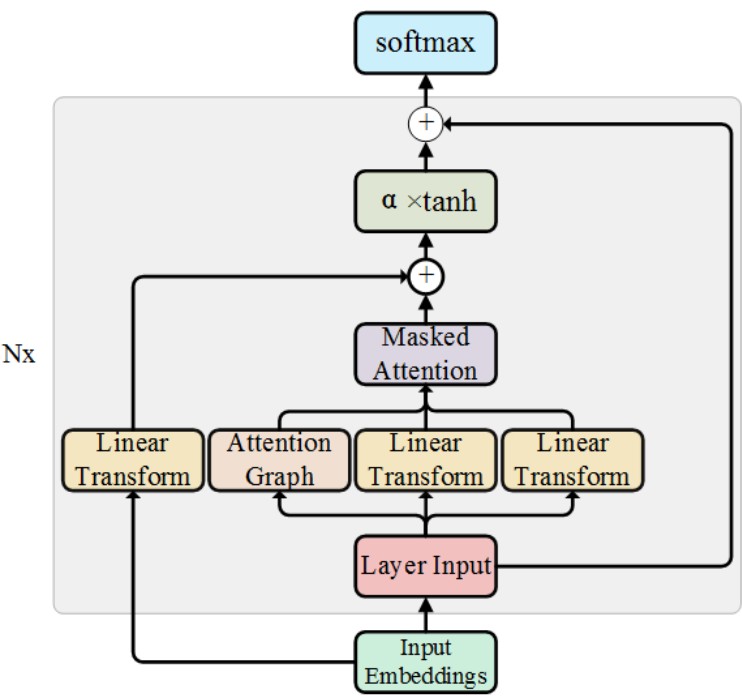

Figure 1: The model structure of a 1layer-Starber

# 4 EXPERIMENTS

## 4.1 A SPECIAL TYPE OF PRETRAINING: EFFECTIVENESS OF MASKED LANGUAGE

Although we don't pretrain on an enormous dataset, we can still improve the performance by utilizing the common pretraining task. In practice, we pretrain our model on a collection of all the inputs without their labels from the training, development and test set. We use dynamic mask (Liu et al., 2019), which is to mask different positions of a sample at every epoch. The reconstruction loss is computed on all positions of a sequence instead of only those masked ones. Masking probability in every position is set to be 0.3. Different from BERT, we don't have a MASK symbol. Instead every masked position is replaced with a word uniformly selected from the whole vocabulary.

Table 1: Model Configurations

|  | Configurations |  | AFQMC | TNEWS | CMNLI |
|---|---|---|---|---|---|
| Word2Vec | #Layer | BiLSTM | 1 | 1 | 1 |
|  |  | ReZero | 4 | 4 | 4 |
|  |  | Transformer | 4 | 4 | 4 |
|  |  | StarSaber | 4 | 4 | 4 |
|  | #Parameter | BiLSTM | 1.45M | 1.45M | 1.45M |
|  |  | ReZero | 3.97M | 3.97M | 3.97M |
|  |  | Transformer | 3.97M | 3.98M | 4.35M |
|  |  | StarSaber | 2.89M | 2.89M | 2.89M |
| Pretrained | #Layer | ReZero | 6 | 6 | 12 |
|  |  | Transformer | 6 | 6 | 12 |
|  |  | StarSaber-1 | 12 | 12 | 24 |
|  |  | StarSaber-2 | 6 | 6 | 12 |
|  | #Parameter | ReZero | 18.19M | 19.81M | 36.37M |
|  |  | Transformer | 18.20M | 19.82M | 36.75M |
|  |  | StarSaber-1 | 13.47M | 15.09M | 27.30M |
|  |  | StarSaber-2 | 7.17M | 8.79M | 14.69M |

## 4.2 SETTINGS

Experiments are conducted on three datasets from Xu et al. (2020), namely the AFQMC(Sentence Similarity) dataset, the TNEWS(Text Classification) dataset and the CMNLI(Natural Language Inference) dataset. For the two text matching tasks, we concatenate the two input sentences with a seperation. The Adam optimizer (Kingma & Ba, 2014) is used. The learning rate is set to 1e-3 in pretraining and 1e-4 in fine-tuning for ReZero and StarSaber. A learning rate of 1e-4 in both pretraining and fine-tuning is applied to the vanilla Transformer. Random seed is set to 10. We compare StarSaber with BiLSTM and two Transformers. For StarSaber and the two Transformers, we have two versions each for every dataset, namely pretrained and not pretrained ones. For LSTM, we only have a not-pretrained version. In such versions, we utilize word embeddings trained from Word2Vec (Mikolov et al., 2013b;a) with data collected from Wikipedia provided by Li et al. (2018) and finetune the embeddings on the pretraining dataset for each task. In training process, we freeze these embeddings. Our BiLSTM concatenates features encoded by two LSTM encoders of opposite directions. In ReZero Layer Normalization (Lei Ba et al., 2016) and the warmup procedure are dropped. Both Transformer and ReZero have 8 heads and take 4 * Hidden-size as the size for the FFN layer. For models using Word2Vec, the hidden size in every layer and the input size are all set to 300. For models pretrained, they are set to 512. Early Stopping is used and the loss function is Cross-Entropy in pretraining but Hinge Loss in training. More details on model configurations are shown in Table 1.[3] All results are submitted online to www.cluebenchmark.com and test labels are not available. Due to the submission limit of 10 times in a month, we cannot try more configuration settings.

## 4.3 RESULTS AND ANALYSIS

Table 2: Dataset Statistics

| Dataset | #Training Sample | #Dev Sample | #Test Sample | #Characters |
|---------|------------------|-------------|--------------|-------------|
| AFQMC | 34334 | 4316 | 3861 | 1.14M |
| TNEWS | 53360 | 10000 | 10000 | 1.63M |
| CMNLI | 391782 | 12426 | 13880 | 22.24M |

Table 3: Accuracy(%) on datasets

| | Models | AFQMC | | TNEWS | | CMNLI | |
|---|--------|-------|------|-------|------|-------|------|
| | | Dev | Test | Dev | Test | Dev | Test |
| Word2Vec | BiLSTM | 69.00 | 69.90 | 51.01 | 51.17 | 58.55 | 67.30 |
| | ReZero | 69.05 | 69.90 | 49.20 | 50.03 | 58.82 | 67.30 |
| | Transformer | 69.00 | 69.90 | 50.23 | 50.70 | 57.75 | 67.30 |
| | StarSaber | 69.05 | 69.96 | 51.22 | 50.85 | 57.69 | 67.30 |
| Pretrained | ReZero | 69.97 | 70.27 | 52.59 | 52.69 | 69.44 | 69.43 |
| | Transformer | 70.32 | 70.24 | 50.78 | 51.64 | **71.14** | 67.30 |
| | StarSaber-1 | **70.85** | **70.68** | **53.25** | **53.41** | 69.67 | **69.43** |
| | StarSaber-2 | 70.18 | 70.22 | 52.28 | 52.55 | 69.44 | 69.43 |

From Table 2 we can see that both AFQMC and TNEWS are datasets of medium size and CMNLI is a larger dataset compared to the other two. AFQMC is a dataset of sentence similarity, represented in a binary classification task. TNEWS is a dataset of text classification which has 15 classes in total. We don't use the keywords provided in order to make comparision with results in Xu et al. (2020). And CMNLI is a Natural Language Inference dataset containing 3 classes for each sample. Our results are shown in Table 3. Results achieved by different large pretrained models are shown in Table 4. It can be seen that none of those large-scale models can achieve astonishing performance. This is due to the construction approach used by CLUE. They use a specific pretrained baseline to select all samples misclassified. Details can be found in https://github.com/CLUEbenchmark/CLUE. Given the results from BiLSTM and StarSaber, it shows that even pretraining on a small dataset with less time and computational power

---

[3]If not stated, parameter numbers are computed with the size of embedding matrices.

can help improve the performance. Pretraining also allows us to use deeper and larger models. The reason why models with Word2Vec are small is that larger models without pretraining can achieve much worse performance, for they indicate greater search space and are harder to optimize.

Table 4: Accuracy for large-scale pretraining models in Xu et al. (2020)

| Models | AFQMC | | TNEWS | | CMNLI | |
|---|---|---|---|---|---|---|
| | Dev | Test | Dev | Test | Dev | Test |
| Bert-base | 74.16 | 73.70 | 56.09 | 56.58 | 79.47 | 79.69 |
| BERT-wwm-ext-base | 73.74 | 74.07 | 56.77 | 56.86 | 80.92 | 80.42 |
| ERNIE-base | 74.88 | 73.83 | 58.24 | 58.33 | 80.37 | 80.29 |
| RoBERTa-large | 73.32 | 74.02 | 57.95 | 57.84 | 82.40 | 81.70 |
| XLNet-mid | 70.73 | 70.50 | 56.09 | 56.24 | 82.21 | 81.25 |
| RoBERTa-wwm-ext | 74.30 | 74.04 | 57.51 | 56.94 | 80.70 | 80.51 |
| RoBERTa-wwm-large-ext | 74.92 | 76.55 | 58.32 | 58.61 | 83.20 | 82.12 |

For AFQMC, all models with Word2Vec in fact output zero for every sample(class labels are zero and one). This may be due to distribution imbalance in such a dataset. Only those pretrained ones can classify a small fraction of samples into the positive class. Thus an improvement of 0.41% is uneasy to achieve. Another insteresting phenomenon appearing in CMNLI is that the gap between the development set and the test set is surprisingly large. For models with Word2Vec, the gap reaches up to 9.61%. For TNEWS, the input sentence is only the title of a passage. In this dataset, StarSaber-1 outperforms ReZero by 0.72% while StarSaber-2 differs from ReZero by only 0.14%.

It can also be seen from the results of AFQMC and TNEWS that stacking more layers in fact helps improve performance for StarSaber. On the dataset of CMNLI, the fact that StarSaber-1 doesn't outperform StarSaber-2 is probably because 12 layers are enough or even redundant for StarSaber. The same logic can be applied to the fact that StarSaber-1 doesn't outperform ReZero. With enough data and enough model complexity, ReZero and Transformer can perform fairly well. Compared to them, StarSaber is more efficient. In all three datasets, it achieves almost the same results as ReZero with the same number of layers, revealing a simple fact that many parts such as Multi-head Attention and the FFN layer are not necessary within our framework. We can drop all these computationally expensive parts.

ReZero is an improved version of Transformer in Vaswani et al. (2017). It adds a trainable weight in front of the Residual Connection and leads to faster convergence. But the better performance of ReZero here doesn't mean it always outperforms Transformer with Layer Normalization, since samples in all these datasets are selected using Transformer-based models. Such a conclusion drawn from these selected data may not hold generally.

## 5 ABLATION STUDIES

### 5.1 EFFECTIVENESS OF GATES

In an RNN, gates allow gradients to flow back to the more distant past. But in our model, there has been a weighted residual connection to solve such a problem, which means that gates' function of adjusting gradient flows is no more important. Here, we incorporate gates in a different way. Formulas can be found in the appendix. Model configurations and results are in Table 5. Numbers of layers, hidden sizes and input sizes are the same as StarSaber-2.

Table 5: Results of StarSaber-gate. Accuracy format:Test acc(Dev acc)

| | AFQMC | TNEWS | CMNLI |
|---|---|---|---|
| #Parameter | 13.47M | 15.09M | 27.29M |
| Accuracy(%) | 70.27(70.62) | 53.56(53.04) | 69.43(69.95) |

We can compare the results here with the results in Table 3. With the number of layers, the hidden size and the input size fixed, gates certainly help improve the performance. But when parameters are

equally many, that is our implementation of StarSaber-1 which has twice the number of layers, gates don't show any superiority. For simplicity and compactness, we drop all gates. We may also want to drop the gates in LSTM and replace them with a weighted residual connection instead, which is simpler and more efficient. And this weight itself can also be parametrized by a simple non-linear function of hidden vectors.

## 5.2 COMPARISION OF TWO WAYS FOR POSITION ENCODING

We conduct experiments on our proposed methods for position encoding. We at first replace the two matrices representing distinct directions with one and add a position embedding matrix made of cosines and sines to the input. The number of parameters is increased by doubling the number of layers. From Table 6, we can observe that after replacing our implementation of position encoding with the PE matrix in Transformer, performance is even worse than StarSaber-2 with less parameters, especially in CMNLI. It means that the PE in Transformer is not consistent with StarSaber. We may give an intuitive explanation: Because of Transformer's FFN layers, the PE matrix added to the input can in fact be recognized in hidden layers. But StarSaber doesn't have a Feed Forward Network, therefore cannot seperate such mixed information.

Table 6: Results of StarSaber implemented with PE. Accuracy format:Test acc(Dev acc)

|  | AFQMC | TNEWS | CMNLI |
|---|---|---|---|
| #Parameter | 11.89M | 13.51M | 24.14M |
| Accuracy(%) | 70.06(70.06) | 52.05(52.13) | 67.30(66.67) |

## 5.3 EFFECTIVENESS OF DIRECT PATHS

Direct paths seem unecessary in StarSaber since we already have a residual connection. However, from the perspective of fixed point, if we drop these direct paths in each layer, the model will finally converge to the same fixed point for whatever inputs. In order to check whether these direct paths are practical or not, we remove all of them and again increase the number of layers to equalize the number of parameters. From the results of CMNLI in Table 7, we can clearly see the benefits they bring.

Table 7: Results of StarSaber without direct paths. Accuracy format:Test acc(Dev acc)

|  | AFQMC | TNEWS | CMNLI |
|---|---|---|---|
| #Parameter | 10.32M | 11.94M | 20.99M |
| Accuracy(%) | 69.75(70.16) | 52.76(52.31) | 67.30(69.00) |

## 6 CONCLUSION

This paper proposes a framework to transform RNN-based models to attention-based ones. With the perspective to view attention as a way to construct a word relation graph, we transform the vanilla RNN to StarSaber, by defining a set of equations. Other variants of RNN can also be transformed in the same way, such as LSTM and GRU discussed above. In this way, we reduce the number of parameters in Transformer by dropping the FFN layer. Experiments on three datasets and the ablation study show the effectiveness of our model and framework.

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

## A  FORMULAS TO INCORPORATE GATES

The formulas to incorporate gates mentioned in the ablation study are listed below:

$$
\begin{aligned}
h_n^{l+1} &= h_n^l + \alpha^l tanh(r_n^l \circ A_n^l + i_n^l \circ (V^l x_n))^4 \\
A_n^l &= \sum_{i<n} G_{ni}^l U^l h_i^l + \sum_{i \geq n} G_{ni}^l Q^l h_i^l \\
r_n^l &= \sigma(W^{rl} A_n^l + V^{rl} x_n) \\
i_n^l &= \sigma(W^{il} A_n^l + V^{il} x_n) \\
G_{ni}^l &= softmax(g^l, -1) = \frac{g_{ni}^l}{\sum_j g_{nj}^l} \\
g_{ni}^l &= \exp(\frac{(h_n^l)^T W^l h_i^l}{\sqrt{d}})
\end{aligned}
\tag{6}
$$

Note that this is also a demonstration of how to transform a recurrence-based model into an attention based model in our framework.

---

[4] $\circ$ denotes the element-wise product.

