# OpenReview forum: "Transforming Recurrent Neural Networks with Attention and Fixed-point Equations"
_ICLR.cc/2021/Conference — Reject_

### Official Review · AnonReviewer4 · 2020-10-20
**Official Blind Review #4**

**Rating:** 3
**Confidence:** 4

**Review:**

Summary:
This paper aims to incorporate the attention mechanism into recurrent neural networks by using fixed point equations. In particular, the authors define a bidirectional RNN with attention by a fixed point equation and then transform it to a variant of the Transformer block. The proposed model StarSaber is shown to be more parameter efficient than the Transformer model and achieve competitive performance on three CLUE datasets.


Pros:
+ This paper proposes a framework of transforming RNNs to Transformer-style models by incorporating the attention mechanism.
+ Several ablation studies are conducted to support the design choices.


Cons:
-	This paper aims to transform the RNN from the perspective of fixed point equations. But in Eq. (4), different weights are assigned to different layers, which means the model is actually iterating different functions at each timestep (layer). Thus, indeed it’s not a fixed point equation and the proposed model is actually a variant of the Transformer model with a modified structure, so the whole story does not make too much sense to me.
-	The difference between the proposed StarSaber block and the Transformer block needs to be highlighted and discussed more clearly in the main content.
-	The dataset and the pretraining task used for experiments are not commonly used and small-scale. It would be more convincing to include experiments on common and larger-scale benchmarks (e.g., SQuAD and GLUE).
-	The paper is not very well-written and is a bit tedious, for example:
    1)	The related work section is not well organized and includes some unnecessary contents like the discussions about pretraining and MLM. Some discussions about related works on implicit deep learning models could be included, e.g., Deep Equilibrium Models [1], Invertible Residual Networks [2], and Implicit Deep Learning [3].
    2)	Eq. (2) – Eq. (4) are three similar sets of equations and rewriting could be avoided.
    3)	The experiment section includes some experimental details that are not very important (e.g., the dataset statistics) which could be moved to the appendix.


Additional Comments/Questions:
1.	As in the StarSaber model, the linear transformation weights for all the future positions and all the past positions are shared ($U^{right}$ and $U^{left}$ respectively). It seems that it incorporates weaker positional information than the positional embedding of the standard Transformer model since it could not directly distinguish between those future positions (and past positions). Therefore, I feel surprised about the ablation study regarding the positional encoding. Could you provide more details about this experiment and explain more about the result?
2.	It would be great to include some comparison of the computational cost of the StarSaber model and the Transformer model.
3.	How do you tune the learning rate for all the models?
4.	Theorem 3.1 is for real-valued function, it would be better to use a vectorized version as the functions considered in the paper are all vector-valued.
5.	Typo: dadtaset - > dataset



[1] https://arxiv.org/abs/1909.01377
[2] https://arxiv.org/abs/1811.00995
[3] https://arxiv.org/abs/1908.06315

---

### Official Review · AnonReviewer2 · 2020-10-21
**There seems a big gap between the proposed RNN transformer with the model used in experiments, also many typos.**

**Rating:** 4
**Confidence:** 4

**Review:**

In this paper, the authors state that they present a framework to transform Recurrent Neural Networks(RNNs) and their variants into self-attention-style models, with an approximation of Banach Fixed-point Theorem. Within this framework, they proposed StarSaber, by solving a set of equations obtained from RNN with Fixed-point Theorem and further approximate it with a Multi-layer Perceptron.

Major comments-----------------------------
1. In equation (3), there is a weight matrix V, while it becomes V^l in equation (4). I am wondering which model is used in reality? Changing V to V^l is irrelevant to the fixed point scheme. I believe that based on the motivation, the weight matrix V should be shared.

2. Many people believe that MLP in transformers is used for universal approximation. Dropping MLP in StarSaber makes me concerned about even approximation capability of the proposed model.

3. I am confused about how the model shown in Figure 1 is related to Equation (5)? Where the masked attention and linear transformers come from based on your derivation in Equations (3)-(5)?

4. All the experiments seem to have been done at quite small scales. I would like to see the experiments on some more challenging NLP tasks, e.g. the experiments in this github repository https://github.com/kimiyoung/transformer-xl, where the baseline code has been provided.

5. I want the authors to report the computational and memory cost of the proposed model over the baselines.

6. The paper claims that weighted residual connection (Bachlechner et al., 2020) which further alleviates the degradation problem. I want to see the evidence to support this claim.



Minor comments------------------------------
1. computation efficency -> computation efficiency.

2. degration -> degradation.

3. dadtaset -> dataset.

4. What is the name of the proposed model? StarSaber or Starber?

5. There are many other typos, which raises my concerns about how serious the authors are in writing this paper and conducting experiments. It makes me feel that the authors did the work in a very short time and wrote the paper very rushly and carelessly.

---

### Official Review · AnonReviewer3 · 2020-10-28
**Review #3**

**Rating:** 4
**Confidence:** 4

**Review:**

The authors propose to make use RNN cell to replace the connection between continuous layers in Transformer. Although the proposed method makes use RNN cell to replace the heavy MLP layer after self-attention. I still think there's no significant difference between vanilla Transformer. The authors also propose to use position aware bi-directional attention mechanism. While it's hard to say whether it's better than multi-head self-attention in Transformer and the authors don't have an ablation study on this. Moreover, I would like to see some model comparisons on more popular datasets such as GLUE, SQuAD, etc.

Cons:
1. It is hard to tell the benefits of the proposed method compared to vanilla Transformer. The motivation for integrating RNN cell is not clear enough for me. If Feed Forward Network is a big concern on speed, there should have some fair comparison on speed.
2. If the authors care more about final performance. It should be evaluated on more popular datasets like GLUE/SQuAD. Pre-training a base model on Wikipedia should not take too much time. I would appreciate it if the authors can have this experiment.
3. The paper needs some further proofread.

Minor comments:
1. improve the computation efficency. (second paragraph in Intro) --> efficiency
2. third paragraph in page 2, in Transformer achieving still competive -->competitive
3. first paragraph in page 3, retraining on a dadtaset --> dataset
4. first paragraph in page 6,we concatenate the two input sentences with a seperation --> separation or special token?

###update###

As no author response, I would like to keep my rating.

---

### Official Review · AnonReviewer1 · 2020-10-30
**The fixed-point idea is fantastic, but this work doesn't seem faithful to that idea**

**Rating:** 5
**Confidence:** 4

**Review:**

Summary:

This paper proposes a way of "transforming" RNNs into attention based models. The basic idea is to simply allow every cell to read the from all other cells instead of just the one immediately preceding.

The authors present the interesting of idea of defining the model as a system of equations to solve instead of as a set of update rules. This approach is interesting but not novel. It was presented in "Deep Equilibrium Models" (Bai et al. (2019)).

The authors of this paper however cite "Deep Equilibrium Models" but then seem to abandon the idea and instead decide to abandon root-finding for the system of equations and go back to a pretty ordinary MLP. They claim that the MLP "approximates" root-finding, but, as far as I can tell, the authors don't even try to check whether the "approximation" they created is actually approaching a solution for the system of equations or is doing something else altogether.

The authors additionally report some gains by having a more complex position encoding and by adding residual connections.

In addition the authors propose a quick pretraining strategy based on masked language modeling on the concatenation of training, dev and test sets from the target tasks. It is not surprising that this helps, since they don't do large scale pretraining, but it's also dangerously close to cheating to include the test set in pretraining, even if the test labels are not available.

Overall the paper is promising and I would like to see more work on this, but right now there is a lot of space for improvement:
* the authors should actually try to solve the system of equations that expresses their model, even if they use a different technique from Bai et al. (2019) -- otherwise just reframe your paper as a modification to the transformer architecture and don't mention the fixed-point approach.
* the authors shouldn't pretrain on the test set.
* experimental results would stronger if they were compared directly to results from other papers. If the authors don't want to run large-scale pretraining they might want to find a low resource setting with competitive results from other papers.

---

### Decision · Program_Chairs · 2021-01-07
**Final Decision**

**Decision:**

Reject

**Comment:**

The paper proposes a new computational block called "StarSaber" which is a self-attention based block derived from RNN fixed-point approximations.

All the 4 reviewers and the authors agree that the work is not ready for publication. While the motivation of the authors is interesting, some reviewers have raised concerns about the validity of the derivation. All the reviewers agree that authors need to clarify the difference between StarSaber and Transformer, compare with Transformer in large scale pre-training experiments, compare the compute speed, do a lot of ablations to validate the design choices.

I recommend the authors to incorporate all the reviewers' comments and make a stronger submission to a future conference!